# Deep Whole-Body Control: Learning a Unified Policy for Manipulation and Locomotion

**Zipeng Fu**[*]     **Xuxin Cheng**[*]     **Deepak Pathak**
Carnegie Mellon University

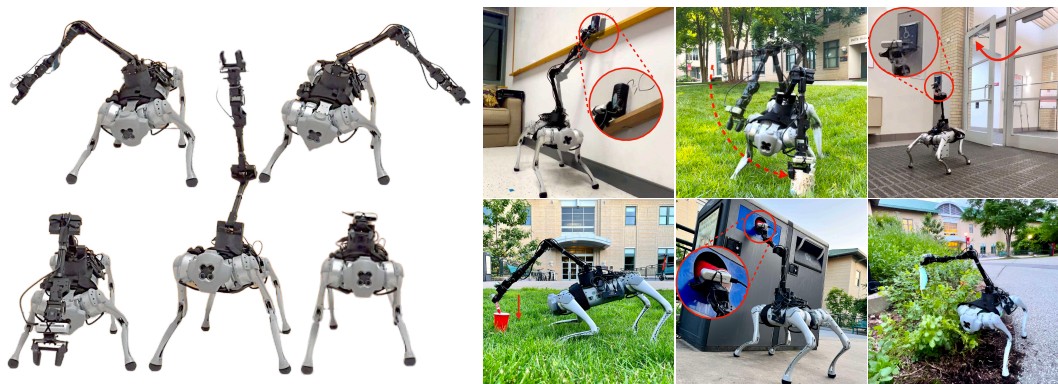

Figure 1: We present a framework for whole-body control of a legged robot with a robot arm attached. Left half shows how whole-body control achieves larger workspace by leg bending and stretching. Right half shows different real-world tasks, including wiping whiteboard, picking up a cup, pressing door-open buttons, placing, throwing a cup into a garbage bin and picking in clustered environments. Videos on the project website.

**Abstract:** An attached arm can significantly increase the applicability of legged robots to several mobile manipulation tasks that are not possible for the wheeled or tracked counterparts. The standard modular control pipeline for such legged manipulators is to decouple the controller into that of manipulation and locomotion. However, this is ineffective. It requires immense engineering to support coordination between the arm and legs, and error can propagate across modules causing non-smooth unnatural motions. It is also biological implausible given evidence for strong motor synergies across limbs. In this work, we propose to learn a unified policy for whole-body control of a legged manipulator using reinforcement learning. We propose Regularized Online Adaptation to bridge the Sim2Real gap for high-DoF control, and Advantage Mixing exploiting the causal dependency in the action space to overcome local minima during training the whole-body system. We also present a simple design for a low-cost legged manipulator, and find that our unified policy can demonstrate dynamic and agile behaviors across several task setups. Videos are at `https://maniploco.github.io`

**Keywords:** Mobile Manipulation, Whole-Body Control, Legged Locomotion

## 1   Introduction

Locomotion has seen impressive performance in the last decade with results in challenging outdoor and indoor terrains, otherwise unreachable by their wheeled or tracked counterparts. However, there are strong limitations to what a legged-only robot can achieve since even the most basic everyday tasks, besides visual inspection, require some form of manipulation. This has led to widespread interest and progress towards building legged manipulators, i.e., robots with both legs and arms, primarily

---

[*]equal contribution; ZF is now at Stanford University.

6th Conference on Robot Learning (CoRL 2022), Auckland, New Zealand.

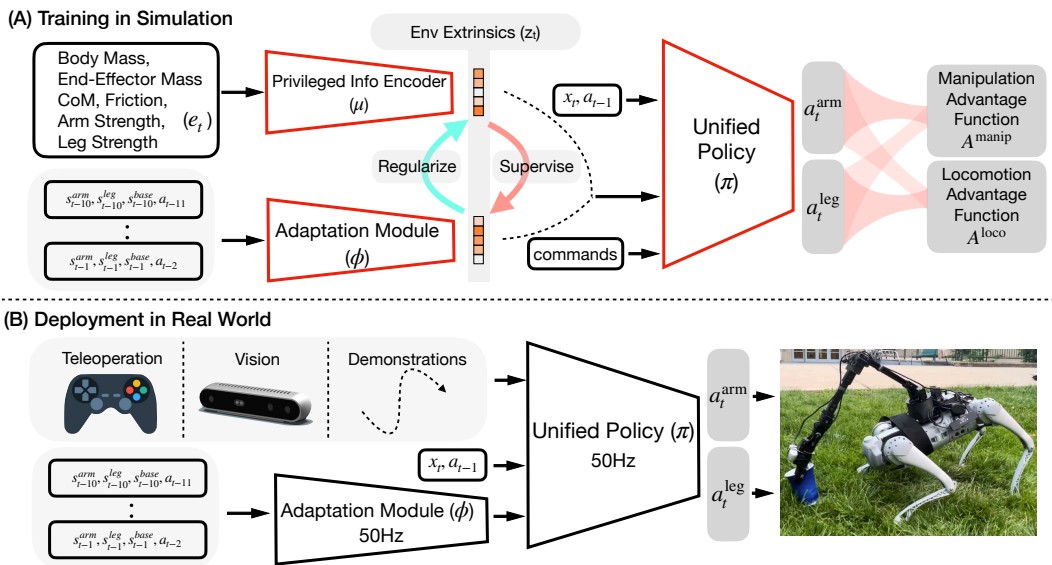

Figure 2: Whole-body control framework. During training, a unified policy is learned by conditioned on environment extrinsics. During deployment, the adaptation module is reused without any real-world fine-tuning. The robot can be commanded in various modes including teleoperation, vision and demonstration replay.

achieved so far through physical modeling of dynamics [1, 2, 3, 4, 5, 6]. However, modeling a legged robot with an attached arm is a dynamic, high-DoF, and non-smooth control problem, requiring substantial domain expertise and engineering effort on the part of the designer. The control frameworks are often modular where kinematic constraints are dealt with separately for different control spaces [7], thus limited to operating in constrained settings with limited generalization. Learning-based methods, such as reinforcement learning (RL), could help lower the engineering burden while aiding generalization to diverse scenarios.

However, recent learning-based approaches for legged mobile manipulators [8] have also followed their model-based counterparts [9, 10] by using modular models in a semi-coupled fashion to control the legs and arm. This is ineffective due to several practical reasons including lack of coordination between the arm and legs, error propagation across modules, and slow, non-smooth and unnatural motions. Furthermore, it is far from the whole-body motor control in humans where studies suggest strong coordination among limbs. In fact, the control of hands and legs is so tied together that they form low-dimension synergies, as outlined over 70 years ago in a seminal series of writings by Russian physiologist Nikolai Bernstein [11, 12, 13]. Perhaps the simplest example is how it is hard for humans to move one arm and the corresponding leg in different motions while standing. The whole-body control should not only allow coordination but also extend the capabilities of the individual parts. For instance, our robot bends or stretches its legs with the movement of the arm to extend the reach of the end-effector as shown in Figure 1.

Unlike legged locomotion, it is not straightforward to scale the standard sim2real RL to whole-body control to several challenges: (a) *High-DoF control*: Our robot shown in Figure 3 has total 19 degrees of freedom. This problem is exacerbated in legged manipulators because the control is dynamic, continuous and high-frequency, which leads to an exponentially large search space even in few seconds of trajectory. (b) *Conflicting objectives and local minima*: Consider when the arm tilts to the right, the robot needs to change the walking gait to account for the weight balance. This curbs the locomotion abilities and makes training prone to learn only one mode (manipulation or locomotion) well. (c) *Dependency*: Consider picking an object on the ground, the end-effector of the arm needs support from the torso by bending legs. This means the absolute performance of manipulation is bounded until legs can adapt.

In this work, we present both a hardware setup for customized low-cost fully untethered legged manipulators and a method for learning one unified policy to control and coordinate both legs and arm, which is compatible with diverse operating modes as shown in Figure 1. We use our unified policy for whole-body control, i.e. to control the joints of the quadruped legs as well as the manipulator to simultaneously take the arm end-effector to desired poses and command the quadruped to move

| | Command Following ($r_{\text{following}}$) | Energy ($r_{\text{energy}}$) | Alive ($r_{\text{alive}}$) |
|---|---|---|---|
| $r^{\text{manip}}$ | $0.5 \cdot e^{-\|[p,o]-[p^{\text{cmd}},o^{\text{cmd}}]\|_1}$ | $-0.004 \cdot \sum_{j \in \text{arm joints}} \lvert \tau_j \dot{q}_j \rvert$ | $0$ |
| $r^{\text{loco}}$ | $-0.5 \cdot \lvert v_x - v_x^{\text{cmd}} \rvert + 0.15 \cdot e^{-\lvert \omega_{\text{yaw}} - \omega_{\text{yaw}}^{\text{cmd}} \rvert}$ | $-0.00005 \cdot \sum_{i \in \text{leg joints}} \lvert \tau_i \dot{q}_i \rvert^2$ | $0.2 + 0.5 \cdot v_x^{\text{cmd}}$ |

Table 1: Both manipulation and locomotion rewards follow: $r_{\text{following}} + r_{\text{energy}} + r_{\text{alive}}$, which encourages command following while penalizes positive mechanical energy consumption to enable smooth motion [17]. Denote forward base linear velocity $v_x$, yaw angular base velocity $\omega_{\text{yaw}}$, torque $\tau$, joint angle velocity $\dot{q}$.

in desired velocities. The key insights of the method are that we can exploit the causal structure in action space with respect to manipulation and locomotion to stabilize and speed up learning, and adding regularization to domain adaptation bridges the gap between simulation with full states and real world with only partial observations.

We perform evaluation on our proposed legged manipulator. Despite immense progress, there exists no easy-to-use legged manipulator for academic labs. Most publicized robot is Spot Arm from Boston Dynamics [14], but the robot comes with pre-designed controllers that cannot be changed. Another example is the ANYmal robot with a custom arm [8] from ANYBotics. Notably, both these hardware setups are expensive (more than 100K USD). We implement a simple design of low-cost legged Go1 robot [15] with low-cost arm on top (hardware costs 6K USD). Our legged manipulator can run fully untethered with modest on-board compute. We show the effectiveness of our learned whole-body controller for teleoperation, vision-guided control as well as open-loop control setup across tasks such as picking objects, throwing garbage, pressing buttons on walls etc. Our robot exhibits **dynamic** and **agile** leg-arm coordinated motions as shown in videos at https://maniploco.github.io.

## 2 Method: A Unified Policy for Coordinated Manipulation and Locomotion

We formulate the unified policy $\pi$ as one neural network where the inputs are current base state $s_t^{\text{base}} \in \mathbb{R}^5$ (row, pitch, and base angular velocities), arm state $s_t^{\text{arm}} \in \mathbb{R}^{12}$ (joint position and velocity of each arm joint), leg state $s_t^{\text{leg}} \in \mathbb{R}^{28}$ (joint position and velocity of each leg joint, and foot contact indicators), last action $a_{t-1} \in \mathbb{R}^{18}$, end-effector position and orientation command $[p^{\text{cmd}}, o^{\text{cmd}}] \in \mathbb{SE}(3)$, base velocity command $[v_x^{\text{cmd}}, \omega_{\text{yaw}}^{\text{cmd}}]$, and environment extrinsics $z_t \in \mathbb{R}^{20}$ (details in Section 2.2). The policy outputs target arm joint position $a_t^{\text{arm}} \in \mathbb{R}^6$ and target leg joint position $a_t^{\text{leg}} \in \mathbb{R}^{12}$, which are subsequently converted to torques using PD controllers. We use joint-space position control for both legs and the arm. As opposed to operational space control of the arm, joint-space control enables learning to avoid self-collision and smaller Sim-to-Real gap, which is also found to be useful in other setups involving multiple robot parts, like bimanual manipulation [16].

We use RL to train our policy $\pi$ by maximizing the discounted expected return $\mathbb{E}_\pi \left[ \sum_{t=0}^{T-1} \gamma^t r_t \right]$, where $r_t$ is the reward at time step $t$, $\gamma$ is the discount factor, and $T$ is the maximum episode length. The reward $r$ is the sum of manipulation reward $r^{\text{manip}}$ and locomotion reward $r^{\text{loco}}$ as shown in Table 1. Notice that we use the second power of energy consumption at each leg joint to encourage both lower average and lower variance across all leg joints. We follow the simple reward design that encourages minimizing energy consumption from [17].

| Command Vars | Training Ranges | Test Ranges |
|---|---|---|
| $v_x^{\text{cmd}}$ | [0, 0.9] | [0.8, 1.0] |
| $\omega_{yaw}^{\text{cmd}}$ | [-1,0, 1.0] | [-1, -.7] & [.7, 1] |
| $l$ | [0.2, 0.7] | [0.6, 0.8] |
| $p$ | $[-2\pi/5, 2\pi/5]$ | $[-2\pi/5, 2\pi/5]$ |
| $y$ | $[-3\pi/5, 3\pi/5]$ | $[-3\pi/5, 3\pi/5]$ |
| $T_{\text{traj}}$ | [1, 3] | [0.5, 1] |

Table 2: Ranges for uniform sampling of command variables

We parameterize the end-effector position command $p^{\text{cmd}}$ in spherical coordinate $(l, p, y)$, where $l$ is the radius of the sphere and $p$ and $y$ are the pitch and yaw angle. The origin of the spherical coordinate system is set at the base of the arm, but independent of torso's height, row and pitch (details in Supplementary). We set the end-effector pose command $p^{\text{cmd}}$ by interpolating between the current end-effector position $p$ and a randomly sampled end-effector position

| Env Params | Training Ranges | Test Ranges |
|---|---|---|
| Base Extra Payload | [-0.5, 3.0] | [5.0, 6.0] |
| End-Effector Payload | [0, 0.1] | [0.2, 0.3] |
| Center of Base Mass | [-0.15, 0.15] | [0.20, 0.20] |
| Arm Motor Strength | [0.7, 1.3] | [0.6, 1.4] |
| Leg Motor Strength | [0.9, 1.1] | [0.7, 1.3] |
| Friction | [0.25, 1.75] | [0.05, 2.5] |

Table 3: Ranges for uniform sampling of environment parameters

$p^{\text{end}}$ every $T_{\text{traj}}$ seconds:

$$p_t^{\text{cmd}} = \frac{t}{T_{\text{traj}}}p + \left(1 - \frac{t}{T_{\text{traj}}}\right)p^{\text{end}}, \ t \in [0, T_{\text{traj}}].$$

$p^{\text{end}}$ is resampled if any $p_t^{\text{cmd}}$ leads to self-collision or collision with the ground. $o^{\text{cmd}}$ is uniformly sampled from $\mathbb{SO}(3)$ space. Table 2 lists the ranges for sampling of all command variables.

## 2.1 Advantage Mixing for Policy Learning

Training a robust policy for a high-DoF robot is hard. In both manipulation and locomotion learning literature, researchers have used curriculum learning to ease the learning process by gradually increasing the difficulty of tasks so that the policy can learn to solve simple tasks first and then tackle difficult tasks [18, 19, 20]. However, most of these works require many manual tunings of a diverse set of the curriculum parameters and careful design of the mechanism for automatic curriculum.

Instead of introducing a large number of curricula on the learning and environment setups, we rely on only one curriculum with only one parameter to expedite the policy learning. Since we know that manipulation tasks are mostly related to the arm actions and locomotion tasks largely depends on leg actions, we can formulate this inductive bias in policy optimization by mixing advantage functions for manipulation and locomotion to speed up policy learning. Formally, for a policy with diagonal Gaussian noise and a sampled transition batch $D$, the training objective with respect to policy's parameters $\theta_\pi$ is

$$J(\theta_\pi) = \frac{1}{|\mathcal{D}|} \sum_{(s_t, a_t) \in \mathcal{D}} \log \pi(a_t^{\text{arm}} \mid s_t)\left(A^{\text{manip}} + \beta A^{\text{loco}}\right) + \log \pi(a_t^{\text{leg}} \mid s_t)\left(\beta A^{\text{manip}} + A^{\text{loco}}\right)$$

$\beta$ is the curriculum parameter that linearly increases from 0 to 1 over timesteps $T_{\text{mix}}$: $\beta = \min(t/T_{\text{mix}}, 1)$. $A^{\text{manip}}$ and $A^{\text{loco}}$ are advantage functions based on $r^{\text{manip}}$ and $r^{\text{loco}}$ respectively. Intuitively, the Advantage Mixing reduces the credit assignment complexity by first attributing difference in manipulation returns to arm actions and difference in locomotion returns to leg actions, and then gradually anneal the weighted advantage sum to encourage learning arm and leg actions that help locomotion and manipulation respectively. We optimize this RL objective by PPO [21].

## 2.2 Regularized Online Adaptation for Sim-to-Real Transfer

Much prior work on Sim-to-Real transfer utilize the two-phase teacher-student scheme to first train a teacher network by RL using privileged information that is only available in simulation, and then the student network using onboard observation history imitates the teacher policy either in explicit action space or latent space [22, 23, 24, 25]. Due to the information gap between the full state available to the teacher network and partial observability of onboard sensories, the teacher network may provide supervision that is impossible for the student network to predict, resulting in a *realizability gap*. This problem is also noted in Embodied Agent community [26]. In addition, the second phase can only start after the convergence of the first phase, yielding extra burdens for both training and deployment.

To tackle the realizability gap and to remove the two-phase pipeline, we propose Regularized Online Adaptation (shown in Figure 2). Concretely, the encoder $\mu$ takes the privileged information $e$ as input and predict an enviornment extrinsics latent $z^\mu$ for the unified policy to adapt its behavior in different environments. The adaptation module $\phi$ estimates the environment extrinsics latent $z^\phi$ by only condition on recent observation history from robot's onboard sensories. We jointly train $\mu$ with the unified policy $\pi$ end-to-end by RL and regularize $z^\mu$ to avoid large deviation from $z^\phi$ estimated by the adaptation module. The adaption module $\phi$ is trained by imitating $z^\mu$ online. We formulate the loss function of the whole learning pipeline with respect to policy's parameters $\theta_\pi$, privileged information encoder's parameters $\theta_\mu$, and adaptation module's parameters $\theta_\phi$ as

$$L(\theta_\pi, \theta_\mu, \theta_\phi) = -J(\theta_\pi, \theta_\mu) + \lambda||z^\mu - \text{sg}[z^\phi]||_2 + ||\text{sg}[z^\mu] - z^\phi||_2 \,,$$

where $J(\theta_\pi, \theta_\mu)$ is the RL objective discussed in Section 2.1, $\text{sg}[\cdot]$ is the stop gradient operator, and $\lambda$ is the Laguagrian multiplier acting as regularization strength. The loss function can be minimized by using dual gradient descent: $\theta_\pi, \theta_\mu \leftarrow \arg\min_{\theta_\pi, \theta_\mu} \mathbb{E}_{(s,a) \sim \pi(\dots, z^\mu)}[L]$,

|  | Survival ↑ | Base Accel. ↓ | Vel Error ↓ | EE Error ↓ | Tot. Energy ↓ |
|---|---|---|---|---|---|
| Unified (Ours) | **97.1** ± 0.61 | **1.00** ± 0.03 | **0.31** ± 0.03 | **0.63** ± 0.02 | 50 ± 0.90 |
| Separate | 92.0 ± 0.90 | 1.40 ± 0.04 | 0.43 ± 0.07 | 0.92 ± 0.10 | 51 ± 0.30 |
| Uncoordinated | 94.9 ± 0.61 | 1.03 ± 0.01 | 0.33 ± 0.01 | 0.73 ± 0.02 | **50** ± 0.28 |

Table 4: Comparison of unified policy with separate policies for legs-arm, and one uncoordinated policy. The unified policy achieves the best performance given same energy consumption. The test ranges are in Table 2.

$\theta_\phi \leftarrow \arg\min_{\theta_\phi} \mathbb{E}_{(s,a) \sim \pi(...,z^\phi)}[L]$, and $\lambda \leftarrow \lambda + \alpha \frac{\partial L}{\partial \lambda}$ with step size $\alpha$. This optimization process is known to converge under mild conditions [27, 28]. In practice, we alternate the optimization process of the unified policy $\pi$ and encoder $\mu$ and the one of adaptation module $\phi$ by a fixed number of gradient steps. $\lambda$ increases from 0 to 1 by a fixed linear scheme. Notice that RMA [22] is a special case of Regularized Online Adaptation, in which the Laguagrian multiplier $\lambda$ is set to be constant zero and the adaptation module $\phi$ starts training only after convergence of the policy $\pi$ and the encoder $\mu$.

**Deployment**    During deployment, the unified policy and adaptation module executes jointly onboard. To specify commands, we develop three interfaces: teleopertion by joysticks, closed-loop control by using RGB tracking, and open-loop reply of human demonstrations. Details are in Section 3.3.

## 3    Experimental Results

### 3.1    Robot System Setup

The robot platform is comprised of a Unitree Go1 quadraped [15] with 12 actuatable DoFs, and a robot arm which is the 6-DoF Interbotix WidowX 250s [29] with a parallel gripper. We mount the arm on top of the quadruped. The RealSense D435 provides RGB visual information and is mounted close to the gripper of WidowX. Both power of Go1 and WidowX are provided by Go1's onboard battery. Neural network inference is also done onboard of Go1. Our robot system uses only onboard computation and power so it is fully untethered.

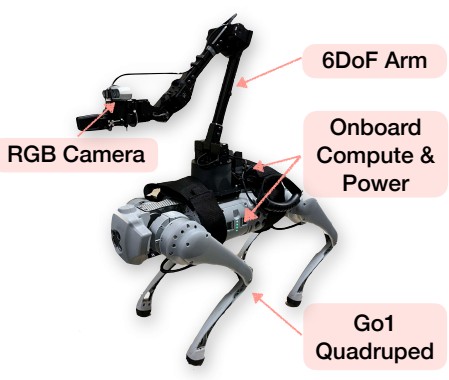

Figure 3: Robot system setup

### 3.2    Simulation Experiments

The purpose of our simulation experiments is to address the following questions:

- Does the unified policy improves over separate policies for the arm and legs? If so, how?
- How Advantage Mixing helps learning the unified policy?
- What's the performance of Regularized Online Adaptation compared with other Sim2Real methods?

**Baselines and Metrics:**    We compare our method with the following baselines:

1. Separate policies for legs and the arm: one policy controls legs based on the quadraped observation, and another policy controls the arm based on arm observation.
2. One uncoordinated policy: Same as unified policy which observes aggregate state of base, legs and the arm, but only $r^{\text{manip}}$ is used to train arm actions, and only $r^{\text{loco}}$ for leg actions.
3. Rapid Motor Adaptation (RMA) [22]: Two-phase teacher-student baseline.
4. Expert policy: the unified policy using the privileged information encoder $z^\mu$.
5. Domain Randomization: the unified policy trained without environment extrinsics $z$.

We report following metrics: (1) survival percentage, (2) Base Accel: angular acceleration of base, (3) Vel Error: L1 error between base velocity commands and actual base velocity, (4) EE Error: L1 error between end-effector (EE) command and actual EE

|  | Arm workspace ($m^3$) ↑ | Survival under perturb ↑ |
|---|---|---|
| Unified (Ours) | **0.82** ± 0.02 | **0.87** ± 0.04 |
| Separate | 0.58 ± 0.10 | 0.64 ± 0.06 |
| Uncoordinated | 0.65 ± 0.02 | 0.77 ± 0.06 |

Table 5: In unified policy, legs help increase the arm workspace and the arm helps the quadruped to stabilize.

pose, (5) Tot. Energy: total energy consumed by legs and the arm. All metrics are normalized by episode length. All experiments are tested over 3 randomly initialized networks and 1000 episodes each. Details of simulation and training are in Supplementary.

| | Realizability Gap $\|z^\mu - z^\phi\|_2$ ↓ | Survival ↑ | Base Accel. ↓ | Vel Error ↓ | EE Error ↓ | Tot. Energy ↓ |
|---|---|---|---|---|---|---|
| Domain Randomization | - | $95.8 \pm 0.2$ | $\mathbf{0.44} \pm 0.00$ | $0.46 \pm 0.00$ | $0.40 \pm 0.00$ | $\mathbf{21.9} \pm 0.53$ |
| RMA [22] | $0.31 \pm 0.01$ | $95.2 \pm 0.2$ | $0.54 \pm 0.02$ | $0.44 \pm 0.00$ | $0.26 \pm 0.04$ | $27.3 \pm 0.95$ |
| Regularized Online Adapt (Ours) | $\mathbf{2e\text{-}4} \pm 0.00$ | $\mathbf{97.4} \pm 0.1$ | $0.51 \pm 0.02$ | $\mathbf{0.39} \pm 0.01$ | $\mathbf{0.21} \pm 0.00$ | $25.9 \pm 0.56$ |
| Expert w/ Reg. | - | $97.8 \pm 0.2$ | $0.52 \pm 0.02$ | $0.40 \pm 0.01$ | $0.21 \pm 0.00$ | $25.8 \pm 0.49$ |
| Expert w/o Reg. | - | $98.3 \pm 0.2$ | $0.51 \pm 0.02$ | $0.39 \pm 0.00$ | $0.21 \pm 0.00$ | $25.6 \pm 0.30$ |

Table 6: Regularized Online Adaptation outperforms other baselines with the smallest imitation error which helps it to have the same performance as the expert policy which uses privileged information to predict environment extrinsics. Expert policy trained with regularization term $\|z^\mu - \mathrm{sg}[z^\phi]\|_2$ has negligible performance degradation compared with the expert trained without regularization. Test ranges in Table 3. Domain Randomization learns to just stand in most cases, hence, trivially collapsing to low Tot. Energy and Base Accel.

**Improvements of the Unified Policy over Baselines:** In Table 4, our unified policy outperforms separate and uncoordinated policies because both the arm and leg actions are trained with the sum of reward for manipulation and locomotion are given with observations for the arm, legs and the quadraped base, while baselines struggle to maintain a small base acceleration, which results in larger error in command velocity following and inaccurate EE pose following.

**Unified Policy Increases Whole-body Coordination:** Table 5 shows that our unified policy promotes whole-body coordination where (1) leg actions will help the arm to achieve a larger workspace via bending for lower EE commands and standing up high for higher EE commands, and (2) arm will help the robot balance under larger perturbation (1.0 m/s initial velocity of base) resulting in higher survival rate of the unified policy. We estimate the the arm workspace via calculating the volume of the convex hull of 1000 sampled EE poses, subtracted by the volume of a cube that encloses the quadruped.

**Advantage Mixing Helps Learning the Unified Policy:** Without Advantage Mixing, the unified policy has difficulty in credit assignment, resulting in the policy first learns EE command following but ignores the locomotion task. As shown in Figure 4, Advantage Mixing helps the policy to focus on each task first and then merge them together, which induces a curriculum-like mechanism to speed up training. Details in Supplementary.

**Robust OOD Performance of Regularized Online Adaptation:** We find that our Regularized Online Adaptation is more robust than RMA and Domain Randomization (DR), tested in environments with out-of-distribution (OOD) environment parameters in Table 3. In RMA, it is not guaranteed the estimated environment extrinsics by the adaptation module can

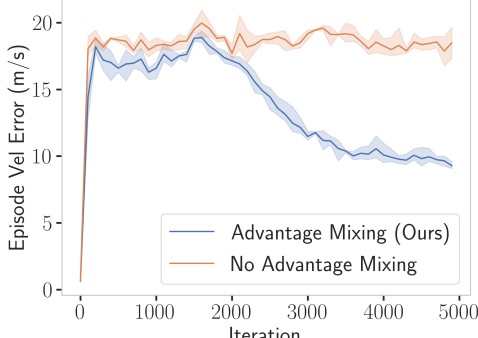

Figure 4: Advantage Mixing helps the unified policy to learn to follow the command velocity much faster (aggregated Vel Error over episodes decreases sharply) than without mixing.

imitates the one learned by the expert. With Regularized Online Adaptation, the expert learns to predict environment extrinsics with regularization from the adaptation module, thus tiny imitation error, **resulting in 20% reduction in EE Error**. Table 6 shows that adding regularization to expert has negligible negative impact on performance, while every metric gets improved compared to RMA due to smaller latent imitation error. Note that DR has better base acceleration and total energy as it just stands in place under difficult environments.

### 3.3 Real-World Experiments

We use the built-in Go1 MPC controller and the IK solver for operational space control of WidowX as the baseline in the real world, which we refer to as MPC+IK. More details are in the Supplementary.

**Teleoperation:** We specify EE position command $p_t^{\mathrm{cmd}}$ by parameterizing $p_{t+1}^{\mathrm{cmd}} = p_t^{\mathrm{cmd}} + \Delta p$, where $\Delta p = (\Delta l, \Delta p, \Delta y)$ is specified by two joysticks. With human in the loop, we can command the end-effector to reach points within or outside of training distribution. In Figure 5, we analyze the whole-body control in the real world, and show that the quadruped's base rotation $(r^{\mathrm{quad}}, p^{\mathrm{quad}}, y^{\mathrm{quad}})$ strongly correlates with the EE position command $p_t^{\mathrm{cmd}}$. This indicates that our unified policy enables whole-body coordination where the leg joints, as well as the arm joints, help reaching.

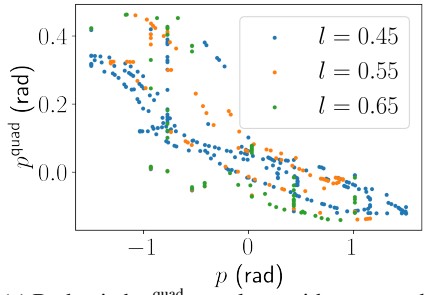
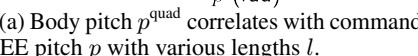
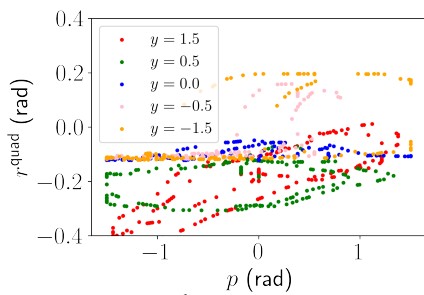

(a) Body pitch $p^{\text{quad}}$ correlates with command EE pitch $p$ with various lengths $l$.

(b) Body pitch $r^{\text{quad}}$ correlates with command EE pitch $p$ with various yaws $y$.

Figure 5: Real-world whole-body control analysis. (a) We fix command EE yaw $y = 0$ and change command EE pitch $p$ and length $l$. When $p$ has a large magnitude, the quadruped will pitch upward or downward to help the arm reach for its goal. With larger $l$ (goal far away), the quadruped will pitch more to help. (b) When the magnitude of command EE yaw $y$ is closer to $1.578$ (arm turns to a side of the torso), the quadruped will roll more to help the arm. When $y = 0$, the quadruped pitches downward instead of roll sideways to help the arm.

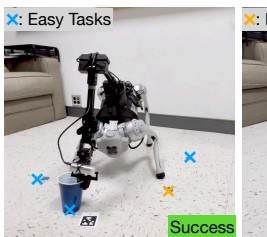
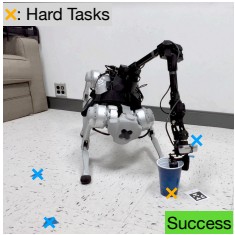
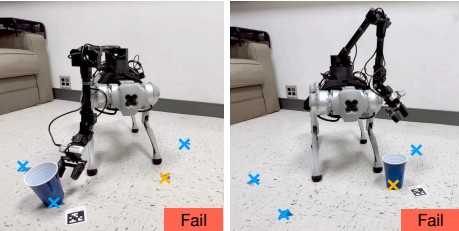

(a) Our method: success in both easy and hard tasks with coordinated behaviors.

(b) MPC+IK: failure in both scenarios. Left: fail due to missed cup. Right: fail due to self-collision.

Figure 6: Comparison of our method and the baseline controller (MPC+IK) in vision-guided pick-up tasks. We sample different points around the robot as the target pick-up position. Easy tasks: 3 points are in normal distance from the robot. Hard tasks: when the point is very close to the front feet and are hard to reach without whole-body control. More hard tasks are in the Supplementary. Videos are at `https://maniploco.github.io`

**Vision-Guided Tracking:** In addition to joystick control by humans, we also show successful picking tasks using visual feedback from an RGB camera. We mount a Realsense D435i camera near the gripper of the arm and use AprilTag [30] to get the relative position between the gripper and the object to be picked up. AprilTag is a visual fiducial system popular in robotics research using simple 2D black and white blocks to encode pose information. We first get the translation of the AprilTag in the camera frame $p^{\text{tag}} = [x^{\text{tag}}, y^{\text{tag}}, z^{\text{tag}}]^T$.

| | Success Rate ↑ | TTC ↓ | IK Failure Rate ↓ | Self-Collision Rate ↓ |
|---|---|---|---|---|
| *Easy tasks (tested on 3 points)* | | | | |
| Ours | **0.8** | **5s** | - | **0** |
| MPC+IK | 0.3 | 17s | 0.4 | 0.3 |
| *Hard tasks (tested on 5 points)* | | | | |
| Ours | **0.8** | **5.6s** | - | **0** |
| MPC+IK | 0.1 | $22.0s$ | 0.2 | 0.5 |

Table 7: Comparison of our method v.s. MPC+IK on pick-up tasks. $p^{\text{end}}$ is the goal position sampled from the points on the ground. TTC is the average time to completion. Each task performance is averaged on 10 real-world trials.

Then we design and use a simple yet effective position feedback controller to set the current EE position command $p_t^{\text{cmd}} = K^T p^{\text{tag}}$, where $K = [-1.5, -1.5, 0.1]^T$ is a gain vector for position control. In Figure 6 and Table 7, we compare our method and the baseline (MPC+IK) in several pick-up tasks by measuring the success rate, average time to to completion (TTC), IK failure rate, and self-collision rate for every setting. We initialize the robot to the same default configuration and before execution.

*Analysis of Success and Failure Modes*: Our method succeeds most of times on easy tasks without visible performance drop in hard task. The failed trials of our method are largely due to the mismatch between the actual cup position and the AprilTag position, which can be mitigated by using two AprilTags and averaging their poses (details in the Supplementary). Since the visual estimation is not the focus of this work, we infer that our method has higher precision and higher efficiency on pick-up tasks than MPC+IK. MPC+IK succeeds in some of the easy tasks and fails due to IK singularity or self-collision. In hard tasks, the major failure cause is self-collision given the cup is too close to the

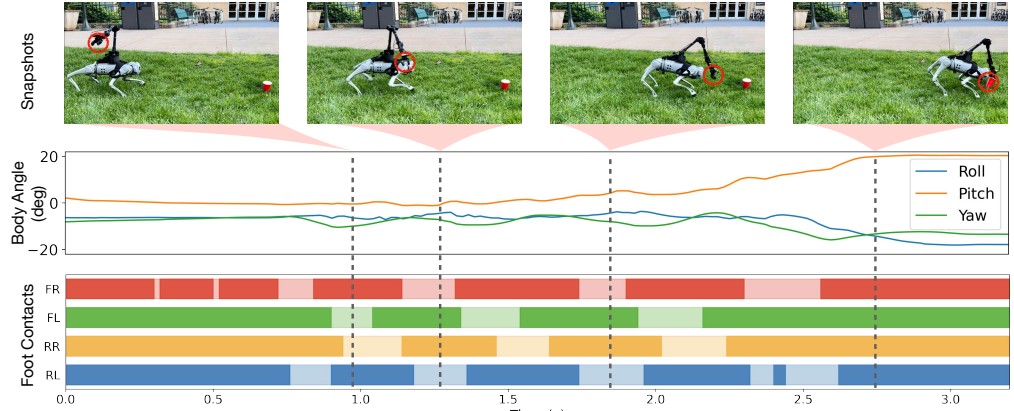

Figure 7: The arm follows a demonstration trajectory to pick up a cup while walking. The start position is $p = (0.5, -0.5, -1.2)$, at the right upper side of the robot and the end position is $p^{\text{end}} = (0.55, -0.9, 0.4)$, on the left lower front side close to the ground. $T_{\text{traj}} = 2.5s$. The robot initially stands on the ground and then is commanded by a constant forward velocity $v_x^{\text{cmd}} = 0.35$. Meanwhile the EE position command changes. When EE position command is high, the quadruped starts to walk without significant tilting behavior with a natural walking gait. As the EE position command moves below the torso, the quadruped starts to pitch downwards, roll to the left and yaw slightly to the right to help the arm reach the goal.

body. Notice that the TTC of MPC+IK is also longer than our method because solving online IK and operational space control more computationally demanding than joint position control (ours).

**Open-loop Control from Demonstration:** In this part, we analyze how agile walking is coupled with dynamic arm movement. The robot is given a pre-defined end-effector trajectory to follow in an open-loop manner while being commanded to walk at the same time. Results in Figure 7 show **agility** and **dynamic coordination** of our legged manipulator on uneven grass terrain powered by our whole-body control method.

## 4   Related Work

**Legged Locomotion**   Traditional model-based control methods for legged robots have shown success but often require controllers to be meticulously designed and many manual tunings [1, 2, 3, 4, 5, 31, 32, 33, 34, 35, 36, 6, 37]. The extra weight and movement of a robot arm on top of the legged robot will make such design process more challenging. Recent advances in reinforcement learning enable legged robots to traverse challenging terrains and adapt to changing dynamics [38, 24, 22, 39, 40, 41, 42, 43, 44, 45, 46, 47, 17, 48, 49, 50]. However these works only focus on the mobility part and few interactions with objects or the environment by manipulation are studied.

**Mobile Manipulation**   Adding mobility to manipulation is studied in [51, 52, 53, 8, 54, 55, 10, 9, 56, 57, 58]. Advances have also been made in the field of biped humanoid [59, 60, 61, 62]. More recently, Ma et al. [8] proposed using an MPC controller to track the desired end-effector position of the arm mounted on a quadruped with a RL policy to maintain balance. However, the controllers for legs (RL) and arm (model-based) are separate modules and no dynamic movements are demonstrated. In [55], language models are used to guide a mobile robot to finish different tasks using the arm. However, the manipulation and mobility are utilized in a decoupled step-by-step manner.

## 5   Discussion and Limitations

We proposed a hardware setup as well as an algorithm to learn whole-body control of a legged robot with robotic arm. Our policy shows coordination between legs and arm while being able to control them in a dynamic manner. Although we have shown preliminary results on object interaction (e.g. picking, pressing, erasing), incorporating general-purpose object interaction (e.g. occlusion and soft object) into the our unified policy is a challenging open research direction. There are several ways in which the current methodology could be extended, such as, learning vision-based policies from the egocentric camera mounted on torso [63] and on the arm, climbing on the obstacle using front legs to pick something up on the table where the arm alone cannot reach, and etc. We believe this paper provides a first step towards several of such future directions.

**Acknowledgments**

We would like to thank Chris Atkeson for high-quality feedback, and Kenny Shaw, Russell Mendonca, Ellis Brown, Heng Yu, Unitree Robotics (Irving Chen, Yunguo Cui and Walter Wen) and staff at CMU Tech Spark for help in real-world experiments, hardware design and assembly. This work is supported in part by DARPA Machine Common Sense grant and ONR N00014-22-1-2096.

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
