# OpenReview forum: "Deep Whole-Body Control: Learning a Unified Policy for Manipulation and Locomotion"
_robot-learning.org/CoRL/2022/Conference — CoRL 2022 Oral_

### Official Review · Reviewer_KgX5 · 2022-07-08

**Originality:** Good
**Technical Quality:** Very Good
**Clarity Of Presentation:** Very Good
**Impact:** 3

**Recommendation:**

Strong Accept: I recommend accepting the paper and will argue for my recommendation even if other reviewers hold a different opinion.

**Summary:**

The authors present a lowcost quadrupped robot with a manipulator. They then introduce a controller that, given commands for the end-effector and base, learns a single policy to control both arm and locomotion to track this command.

**Issues:**

- l65: what does low-cost mean in comparison to the 100k for ANYmal?
- l141: "This optimization process is known to converge." I believe convergence is only guarantueed under certain assumptions.
- l166: Why not give the "separate" baseline access to the same observations, i.e. both base and arm state?
- Minor: there are a number of spelling/grammar mistakes that could be cleared up
- l200: This section should make clear what is OOD. At the moment the reference to table 3 is hidden in the caption.
- l239: TTC is never explained in the text
- Table 7: would be beneficial to not just have 1 difficult point
- l 251: grouping 17 references into a single parantheses seems a bit much
- Sec. 5: obstacle avoidance? As I understand it, the current policy is blind?

**Quality Of The Limitations Section:**

Additional details required

**Reviewer Expertise:**

3: The reviewer is fairly confident that the evaluation is correct

**Robotics Focus:**

Sufficient demonstration on hardware

**Strengths And Weaknesses:**

Strengths
- the advantage mixing approach seems simple and sensible for this domain
- the regularization during the teaching process seems to work well
- the experiments and videos demonstrate good behavior and the usefullness of the approach

Weaknesses
- the paper does not make fully clear from the beginning that the task is to follow given commands. E.g. l76 p^cmd, v^cmd are just appearing as given. It would be beneficial for clarity to describe that the task is to follow given commands earlier on.
- the paper proposes to jointly train teacher and student instead of in two phases, but it does not evaluate the impact of this. It would be good to compare to the "standard" approach of training in two phases.
- generally the experiments could benefit from a few more additions, comparisons, see list of issues below


**Summary Of Recommendation:**

Overall a good paper. Though additional comparison and a bit more clarity in certain parts would be very beneficial.

Post rebuttal: the authors have addressed my main concerns, with the changes discussed in the process I think the paper is ready to be accepted and I am happy to raise my score

---

> ### Author Response · Authors · 2022-08-20
> **Response to Reviewer # KgX5 (with new experiments) [part 1/2]**
>
> Dear Reviewer,
>
> Thank you for the insightful feedback! We provide clarifications for your concerns below and are also happy to report that we finished the experiments you suggested for the hard visual tracking task. We also included another experiment that Reviewer bkQw suggested. We hope that these answers (+ new experiments) address your concerns, and if so, we request the reviewer to consider updating the score. Otherwise, please let us know any further concerns that remain.
>
> > *the paper proposes to jointly train teacher and student instead of in two phases, but it does not evaluate the impact of this. It would be good to compare to the "standard" approach of training in two phases.*
> - Sorry, it might have gotten missed but we evaluated the impact of jointly training teacher and student instead of in two phases in Table 6 and Section 3.2 “Robust OOD Performance of Regularized Online Adaptation''. The RMA [22] baseline is trained in two phases, as we mentioned in L144 & 145.
> - From results, it is clear that RMA performs worse due to a much larger imitation error of the student policy, resulting in a **24% increase in end-effector command following error** in student policy. However, with our method (Regularized Online Adaptation), the expert learns to predict environment extrinsics with regularization from the adaptation module, thus resulting in **tiny imitation error** of our student policy. We also compared the expert policies in two-phase training (*Expert w/o Reg.*) and in our one-phase training (*Expert w/ Reg.*), and showed that no significant difference in performance was observed.
> - We will make it further clear in the camera-ready version that RMA [22] is the baseline of two-phase training.
>
> > *l166: Why not give the "separate" baseline access to the same observations, i.e. both base and arm state?*
> - Indeed, this is a different baseline which we already have in the paper under the name “Uncoordinated”. It contains individual policies but both have access to the joint observation of both base and arm state. We define these baselines, “Uncoordinated” as well as “Separate”, in the Section 3.2 “Baselines and Metrics”.
> - We have compared both these baselines with ours (“Unified”) in Table 5 and Section 3.2 “Improvements of the Unified Policy over Baselines”. We showed that the “Uncoordinated” baseline (which reviewer is suggesting) performs better than “Separate” due to its access to both arm and leg states, but both baselines are significantly outperformed by our method “Unified”.
>
> > *Table 7: would be beneficial to not just have 1 difficult point*
> - Thank you.  As per your suggestion, we added new experiments to include 4 more difficult settings (i.e. target points) for the hard visual tracking instead of one in the paper (note that there are already 10 trials per point).
> - One of these 4 points is far away from the robot. Even though MPC+IK is not able to reach it with limited workspace, our method can successfully grasp the cup far away because workspace gets increased due to joint coordination.
> - Other 3 points contain points around the robot that are closer to the robot body that require whole-body coordination to bend the feet for the arm gripper to reach.
> - Result videos: [https://drive.google.com/file/d/1-w8WTo4WISpMAaMniUBjs7YqsvTO3FLx/view?usp=sharing](https://drive.google.com/file/d/1-w8WTo4WISpMAaMniUBjs7YqsvTO3FLx/view?usp=sharing).
> - For each of thes 4 new settings (points), we tested 10 trials with the same metric as in Table 7 in the paper as follows:
>
> | Hard Ground Points ($p^\mathrm{end}$)| Methods | Success Rate ($\uparrow$) | TTC ($\downarrow$) | IK Failure Rate ($\downarrow$) | Self Collision Rate ($\downarrow$) |
> | - | - | - | - | - | - |
> | (0.72, -0.51, 0.34), (0.55, -0.75, -0.43), (0.56, -0.73, 0.5), (0.45, -0.74, 1.80), (0.45, -0.76, -1.8) | Ours | 0.8 | 5.6 | \- | 0 |
> | ^ | MPC+IK| 0.1 | 22 | 0.2 | 0.5 |
>
>
> > *l65: what does low-cost mean in comparison to the 100k for ANYmal?*
> - “Low-cost” means the hardware cost of custom-built our setup is low, and hence our setup is much more accessible compared with legged manipulators. Our quadruped platform (Unitree Go1) only costs 2.7K USD, and the robotic arm (Interbotix WidowX 250s) only costs 3K USD.
> - The downside of low-cost robots includes imprecise behavior and less repeatability which we address through our Regularized Online Adaptation.
> - We have described the hardware setup in detail in the Section F of the supplementary and will add more details about low-cost hardware in the supplementary of the camera-ready version.
>
>
> [1/2] ... continued below ...

---

> > ### Author Response · Authors · 2022-08-20
> > **Response to Reviewer # KgX5 (with new experiments) [part 2/2]**
> >
> > > *the paper does not make fully clear from the beginning that the task is to follow given commands. E.g. l76 p^cmd, v^cmd are just appearing as given. It would be beneficial for clarity to describe that the task is to follow given commands earlier on.*
> > - Thanks for pointing this out. In the camera ready version, in the beginning, we will restate the problem statement of whole-body control, i.e. to control the joint angles of the quadruped as well as the arm through a unified policy so as to simultaneously take the end-effector to desired poses and command the quadruped to move in desired velocities with arm on top.
> >
> > > *l141: "This optimization process is known to converge." I believe convergence is only guarantueed under certain assumptions.*
> > - Thanks for pointing this out. It is true that Bregman ADMM is proven to converge under mild assumptions. We will add this clarification in the camera-ready version.
> >
> > > *l200: This section should make clear what is OOD. At the moment the reference to table 3 is hidden in the caption.*
> > - We will mention in the camera-ready version that OOD means out-of-distribution.
> >
> > > *l239: TTC is never explained in the text*
> > - We defined the abbreviation TTC (average time to completion) in the caption of Table 7, where this metric is first used.
> > - In the camera ready version, we will define TTC earlier as well.
> >
> > > *l 251: grouping 17 references into a single parentheses seems a bit much*
> > - Thanks for this suggestion. We will modify the camera-ready version accordingly.
> >
> > > *Sec. 5: obstacle avoidance? As I understand it, the current policy is blind?*
> > - Yes, the focus of the paper is on whole body control from proprioceptive input. In this work, we condition on the visual information only through low-dimensional commands.
> > - Therefore, we mentioned in the *Section 5 Limitations* that, in the future, researchers can train visual control policies directly from the camera on the arm, climbing on the obstacle using front legs to pick something up on the table where the arm can not reach, and etc. And we hope that this paper provides a first step towards several of such future directions.
> >
> > $\newline$
> >
> > Below we report another experiment we did for the rebuttal upon request from other reviewers. We hope you find them useful as it further justifies our approach.
> >
> > ### New experiment: more accurate visual tracking by averaging poses of two April Tags
> > - We tested the 2-tag method proposed by Reviewer bkQw to capture the track more reliably so that the results can purely focus on the whole-body control performance. We will add this into the camera ready. The result video is here: [https://drive.google.com/file/d/1QsgC7horF8xsTwuBFOO3Z7zujCkepTj3/view?usp=sharing](https://drive.google.com/file/d/1QsgC7horF8xsTwuBFOO3Z7zujCkepTj3/view?usp=sharing)
> >
> >
> > [2/2]

---

> > > ### Comment · Reviewer_KgX5 · 2022-08-24
> > > **Response**
> > >
> > > I would like to thanks the authors for their response, which has clarified my main concerns. Small notes:
> > >
> > > > Sorry, it might have gotten missed but we evaluated the impact of jointly training teacher and student instead of in two phases in Table 6 and Section 3.2 “Robust OOD Performance of Regularized Online Adaptation''. The RMA [22] baseline is trained in two phases, as we mentioned in L144 & 145.
> > >
> > > Please do make this more clear. Line 170 currently states the opposite by saying it is the same except for lambda
> > >
> > > > We will mention in the camera-ready version that OOD means out-of-distribution.
> > >
> > > I was not refering to explain the abbreviation (though that would be good as well), but rather to mention which aspect of the problem is out-of-distribution

---

> > > > ### Author Response · Authors · 2022-08-24
> > > > **Follow-up Response to Reviewer # KgX5**
> > > >
> > > > Dear Reviewer,
> > > >
> > > > Thank you for pointing out that we should mention which aspect of the problem is out-of-distribution! Section “Improvements of the Unified Policy over Baselines” and Table 4 are tested using OOD ranges in Table 2 for the end-effector and base commands (e.g. command end-effector positions being further away from the base, etc). Section “Robust OOD Performance of Regularized Online Adaptation” and Table 6 are tested using OOD ranges in Table 3 for environment parameters (e.g. slipperier terrain). We will include this and other rebuttal answers in the camera ready.
> > > >
> > > > Please let us know of any remaining questions. If your concerns are addressed, we kindly request whether the reviewer could consider updating the recommendation score. Thank you once again!

---

> > > > > ### Author Response · Authors · 2022-08-28
> > > > > **Final follow-up response**
> > > > >
> > > > > Dear Reviewer,
> > > > >
> > > > > Today is the end of the discussion period. We hope that our answers have addressed all your concerns. If so, we kindly request whether the reviewer could consider updating the confidence or the final recommendation. Please let us know if there are other comments for the camera ready. Thank you once again!

---

### Official Review · Reviewer_bkQw · 2022-07-29

**Originality:** Very Good
**Technical Quality:** Very Good
**Clarity Of Presentation:** Good
**Impact:** 3

**Recommendation:**

Strong Accept: I recommend accepting the paper and will argue for my recommendation even if other reviewers hold a different opinion.

**Summary:**

This paper presents a method for learning a coordinated policy to control a quadruped with mounted arm. There are two key algorithmic components. The first is advantage mixing, wherein the locomotion and manipulation policies are first trained independently, then gradually integrated together with shared reward terms. The second key algorithmic component is Regularized Online Adaptation, a modification of RMA where the environment encoder with privileged information is updated throughout training to more closely match the adaptation module, which has no privileged information. The resulting whole-body controller rolls and pitches the body to help the arm reach, as well as successfully trotting and reaching. This is demonstrated on hardware in three control scenarios.

**Issues:**

-  Section 2.2 should be clarified. In particular it would benefit from a brief summary of RMA and a more thorough explanation of the differences between Regularized Online Adaptation and RMA.
-  From line 93, I understood that the spherical coordinate system for the end effector was relative to the robot's torso, which would mean the legs could not help the arm reach. From the supplementary material I realize that the coordinate system is actually independent of the torso's height, roll, and pitch. This should be stated in the main paper.

Typos:
-  Line 2, "picking in [cluttered] environments."
-  Line 63, "pre-designed controllers that [can't] be changed."
-  Line 73, "([roll], pitch, and base angular velocities)"
-  Line 139, "[Lagrangian] multiplier"
-  Line 148, "open-loop [replay] of human demonstrations"


**Quality Of The Limitations Section:**

Limitations are addressed clearly

**Reviewer Expertise:**

3: The reviewer is fairly confident that the evaluation is correct

**Robotics Focus:**

Sufficient demonstration on hardware

**Strengths And Weaknesses:**

Strengths:
-  Very nice results. It's awesome to see the torso tilt to stretch the arm farther, especially in the video.
-  Techniques are relatively simple, but they seem to make a big difference in the learned policy.
Thorough ablation studies.
-  I like the idea of Regularized Online Adaptation. I wasn't convinced at first, but after thinking about it, here is my reasoning: Suppose there is some privileged information the encoder is encoding that the adaptation module simply cannot infer from the information it has. The policy may come to rely on that information, but the adaptation module cannot provide it. By training the encoder to match the adaptation module, you ensure the encoder isn't passing on any information the adaptation module cannot infer.

Weaknesses:
-  Section 2.2 (Regularized Online Adaptation for Sim-to-Real Transfer) could be clearer. In particular it would benefit from a brief summary of RMA and a more thorough explanation of the differences between Regularized Online Adaptation and RMA.
-  Line 234 says "The failed trials of our method are largely due to the mismatch between the actual cup position and the AR tag position." I understand that vision is not the focus of this work, but as this comparison with MPC+IK is an important result, this visual estimation issue should be resolved. As one simple option, why not put an April tag on each side of the cup and average the coordinates?


**Summary Of Recommendation:**

There are a couple places where the paper should be more clear, but overall it is convincing. The problem addressed is an interesting one: yes, robots with both arms and legs should ideally coordinate their movements, as humans do. The two non-standard algorithmic components (advantage mixing and regularized online adaptation) make sense and are well-investigated with ablations and comparison methods. And the results look good: the robot twists its torso to help the arm reach.

---

> ### Author Response · Authors · 2022-08-20
> **Response to Reviewer # bkQw (with new experiments)**
>
> Dear Reviewer,
>
> Thank you for the insightful feedback! We provide clarifications for your concerns below and also happy to report that we finished the new experiment on the visual tracking task by averaging the poses of two April Tags that you suggested. We also included more experiments on the hard visual tracking task that were suggested by Reviewer KgX5.
>
> We hope that these answers (+ new experiments) would help increase your confidence. Please let us know if there are any more questions.
>
> > *Line 234 says "The failed trials of our method are largely due to the mismatch between the actual cup position and the AR tag position." I understand that vision is not the focus of this work, but as this comparison with MPC+IK is an important result, this visual estimation issue should be resolved. As one simple option, why not put an April tag on each side of the cup and average the coordinates?*
> - Thanks a lot for your proposed solution of using two April Tags, one on each side of the target object, and then average the coordinates for better pose estimation! We are glad to report that we tested the 2-tag method and agree that this is a very viable solution. We will consider replacing our 1-tag method to the 2-tag method for the camera ready version.
> - Result video: [https://drive.google.com/file/d/1QsgC7horF8xsTwuBFOO3Z7zujCkepTj3/view?usp=sharing](https://drive.google.com/file/d/1QsgC7horF8xsTwuBFOO3Z7zujCkepTj3/view?usp=sharing).
>
> > *Section 2.2 (Regularized Online Adaptation for Sim-to-Real Transfer) could be clearer. In particular it would benefit from a brief summary of RMA and a more thorough explanation of the differences between Regularized Online Adaptation and RMA.*
> - Thanks for pointing this out! We will add a brief summary of RMA and differences in the camera ready.
> - We would like to also note that although we instantiated Regularized Online Adaptation in the context of RMA [22], it is generally applicable with small changes to other two-phase policy training methods using the Teacher-Student training pipeline [23, 24, 25]. By training the teacher policy with regularization (on action space for [23, 24, 25]) from the student policy in an online fashion, the teacher policy will not learn information that is not learnable by the student policy which can only observe partial onboard information. Hence, the student policy can perform as well as the teacher policy without drop in performance that is common in two-phase policy training [22, 23, 24, 25] due to imitation gap, shown in Section 3.2 Table 6.
>
> > “From line 93, I understood that the spherical coordinate system for the end effector was relative to the robot's torso, which would mean the legs could not help the arm reach. From the supplementary material I realize that the coordinate system is actually independent of the torso's height, roll, and pitch. This should be stated in the main paper.”
> - Thanks a lot for pointing this out! Indeed, it will be clearer if we move this from supplementary to the main paper. We will do so in the camera-ready version.
>
> $\newline$
>
> Below we report another experiment we did for the rebuttal upon request from other reviewers. We hope you find them useful as it further justifies our approach.
>
> ### More experiments for 'hard' visual tracking setting
> - We added new experiments suggested by Reviewer KgX5 to include 4 more difficult settings (i.e. target points) for the hard visual tracking instead of one in the paper (note that there are already 10 trials per point).
> - One of these 4 points is far away from the robot. Even though MPC+IK is not able to reach it with limited workspace, our method can successfully grasp the cup far away because workspace gets increased due to joint coordination.
> - Other 3 points contain points around the robot that are closer to the robot body that require whole-body coordination to bend the feet for the arm gripper to reach.
> - Result videos: [https://drive.google.com/file/d/1-w8WTo4WISpMAaMniUBjs7YqsvTO3FLx/view?usp=sharing](https://drive.google.com/file/d/1-w8WTo4WISpMAaMniUBjs7YqsvTO3FLx/view?usp=sharing).
> - For each of thes 4 new settings (points), we tested 10 trials with the same metric as in Table 7 in the paper as follows:
>
> | Hard Ground Points ($p^\mathrm{end}$)| Methods | Success Rate ($\uparrow$) | TTC ($\downarrow$) | IK Failure Rate ($\downarrow$) | Self Collision Rate ($\downarrow$) |
> | - | - | - | - | - | - |
> | (0.72, -0.51, 0.34), (0.55, -0.75, -0.43), (0.56, -0.73, 0.5), (0.45, -0.74, 1.80), (0.45, -0.76, -1.8) | Ours | 0.8 | 5.6 | \- | 0 |
> | ^ | MPC+IK| 0.1 | 22 | 0.2 | 0.5 |
>
> $\newline$
>
> #### Reference
> [22] A. Kumar, et al. RMA: Rapid Motor Adaptation for Legged Robots. RSS, 2021.
> [23] G. B. Margolis, et al. Learning to jump from pixels. CoRL, 2021.
> [24] J. Lee, et al. Learning quadrupedal locomotion over challenging terrain. Sci Robotics, 2020.
> [25] T. Miki, et al. Learning robust perceptive locomotion for quadrupedal robots in the wild. Sci Robotics, 2022.

---

> > ### Comment · Reviewer_bkQw · 2022-08-26
> > **Response**
> >
> > Thank you, authors, for your thorough response. I'm happy to see that averaging two April tags helped with position estimation. I've no further concerns to note.

---

### Official Review · Reviewer_Hcpy · 2022-07-30

**Originality:** Very Good
**Technical Quality:** Very Good
**Clarity Of Presentation:** Very Good
**Impact:** 4

**Recommendation:**

Strong Accept: I recommend accepting the paper and will argue for my recommendation even if other reviewers hold a different opinion.

**Summary:**

The paper presents a novel method for whole body control of a legged manipulator using sim-to-real reinforcement learning. The method is based on a teacher-student framework similar to [1], where an encoder uses privileged information in simulation to predict latent environment extrinsics and an adaptation module learns to mimic it from a history of sensor measurements. The output of the adaptation module is fed into a whole body control policy, and the entire system is trained end-to-end. In contrast to [1], the encoder output is regularized to overcome the realizability gap, and the system is trained in a single phase. The authors also propose advantage mixing to provide a natural curriculum for policy learning. Extensive simulation and hardware experiments demonstrate that the learned controller can successfully coordinate the quadrupedal base and manipulator to perform a variety of complex tasks.

[1] A. Kumar, et. al., RMA: Rapid Motor Adaptation for Legged Robots, RSS 2021.


**Issues:**

The paper draft can be significantly improved from a proof-reading pass to improve several grammatical errors. A few examples,

Line 107 - “rely on” instead of “relies on”
Line 127 - “start” instead of “starts”

The authors should fix these in the final version and address the comments in the weaknesses section.


**Quality Of The Limitations Section:**

Additional details required

**Reviewer Expertise:**

4: The reviewer is confident but not absolutely certain that the evaluation is correct

**Robotics Focus:**

Sufficient demonstration on hardware

**Strengths And Weaknesses:**

**Strengths**

Comprehensive Experimental Evaluation

The real-world robot experiments are an impressive demonstration of learning based control working on complex tasks that require coordination between large degrees of freedom as well as generating dynamic motions. The simulated experiments and baseline comparisons clearly demonstrate the benefits of every of the proposed training methodology.

Training

The paper provides novel contributions to the training strategy. Regularizing the encoder output to overcome the realizability gap allows training the encoder and adaptation module in a single phase, which helps reduce the overall training time. Furthermore, the idea of advantage mixing to exploit the causal structure of the locomotion and manipulation is a simple yet effective technique for avoiding local minima.

**Weaknesses**

1. It would be helpful if the authors discuss more details of the low-cost hardware setup and how it can be re-created by other researchers.

2. The limitations section does not provide any insights into scenarios where the current method will fail and how it can be improved. In particular, the authors should discuss the limits of sim-to-real when it comes to both locomotion and manipulation (especially cases that involve object interaction that can be hard to simulate), and how the current methodology can be potentially adapted to them.


**Summary Of Recommendation:**

My recommendation for strong accept is based on the impressive real-world results, a thorough evaluation of different components and interesting improvements in the training pipeline. In the future it would be interesting to see how such approaches can be used to autonomously perform more complicated and fine-grained manipulation tasks.

---

> ### Author Response · Authors · 2022-08-20
> **Response to Reviewer # Hcpy (with new experiments)**
>
> Dear Reviewer,
>
> Thank you for the insightful feedback! We provide clarifications for your concerns below and include new experiments as suggested by other two reviewers. We hope that these answers (+ new experiments) would help increase your confidence. Please let us know if there are any more questions.
>
> > *It would be helpful if the authors discuss more details of the low-cost hardware setup and how it can be re-created by other researchers.*
> - We could not provide all details in the Section 3.1 Robot System Setup of the main paper due to space issues. Therefore, we had included more thorough details in the Section F of the supplementary material. To reiterate, the supplementary describes the hardware setup as well as the software details to ensure smooth communication between the arm and the quadruped. We will refer to supplementary details clearly in the main paper and take another pass to include even more details in the supplementary of the camera-ready version.
>
> > *The limitations section does not provide any insights into scenarios where the current method will fail and how it can be improved. In particular, the authors should discuss the limits of sim-to-real when it comes to both locomotion and manipulation (especially cases that involve object interaction that can be hard to simulate), and how the current methodology can be potentially adapted to them.*
> - This is a great suggestion, thank you! You are right that object interaction is a challenging setup for sim2real and we will add the following discussion to the paper.
> - Although the focus of this paper is whole-body control (agent-centric), we have still shown some preliminary results on object interaction (environment-centric). However, incorporating general-purpose object interaction into the agent-centric policy is a challenging open research problem particularly because it is unclear how to simulate a diverse variety of contact-rich object interactions, unlike walking terrains that we procedurally generate. Furthermore, object grasping and interaction in occluded scenes and soft object manipulation adds to the difficulty.
> - That being said, some of the learning challenges are similar to our setup as diverse object interaction with a legged manipulator also has a high-dimensional input and output space. Therefore, our proposed ideas of advantage mixing and Regularized Online Adaptation could still be beneficial in circumventing local minimas and for more efficient transfer from simulation to real, respectively, as they do so in whole-body control.
>
> > *The paper draft can be significantly improved from a proof-reading pass to improve several grammatical errors. A few examples,*
> - Thank you for the suggestion! We promise to take a thorough pass and address all the grammatical errors in the camera-ready version.
>
> $\newline$
>
> Below are the new experiments we did for rebuttal upon request from other reviewers. We hope you find them useful as they further justify our approach.
>
> ### New experiment: more accurate visual tracking by averaging poses of two April Tags
> - We tested the 2-tag method proposed by Reviewer bkQw to capture the track more reliably so that the results can purely focus on the whole-body control performance. We will add this into the camera ready. The result video is here: [https://drive.google.com/file/d/1QsgC7horF8xsTwuBFOO3Z7zujCkepTj3/view?usp=sharing](https://drive.google.com/file/d/1QsgC7horF8xsTwuBFOO3Z7zujCkepTj3/view?usp=sharing)
>
> ### More experiments for 'hard' visual tracking setting
> - We added new experiments suggested by Reviewer KgX5 to include 4 more difficult settings (i.e. target points) for the hard visual tracking instead of one in the paper (note that there are already 10 trials per point).
> - One of these 4 points is far away from the robot. Even though MPC+IK is not able to reach it with limited workspace, our method can successfully grasp the cup far away because workspace gets increased due to joint coordination.
> - Other 3 points contain points around the robot that are closer to the robot body that require whole-body coordination to bend the feet for the arm gripper to reach.
> - Result videos: [https://drive.google.com/file/d/1-w8WTo4WISpMAaMniUBjs7YqsvTO3FLx/view?usp=sharing](https://drive.google.com/file/d/1-w8WTo4WISpMAaMniUBjs7YqsvTO3FLx/view?usp=sharing).
> - For each of thes 4 new settings (points), we tested 10 trials with the same metric as in Table 7 in the paper as follows:
>
> | Hard Ground Points ($p^\mathrm{end}$)| Methods | Success Rate ($\uparrow$) | TTC ($\downarrow$) | IK Failure Rate ($\downarrow$) | Self Collision Rate ($\downarrow$) |
> | - | - | - | - | - | - |
> | (0.72, -0.51, 0.34), (0.55, -0.75, -0.43), (0.56, -0.73, 0.5), (0.45, -0.74, 1.80), (0.45, -0.76, -1.8) | Ours | 0.8 | 5.6 | \- | 0 |
> | ^ | MPC+IK| 0.1 | 22 | 0.2 | 0.5 |

---

> > ### Comment · Reviewer_Hcpy · 2022-08-27
> > **Response to Authors**
> >
> > I would like to thank the authors for their clarifications and additional experiments. It would be great to see the discussed changes in the updated draft.

---

### Official Review · Reviewer_3ot8 · 2022-08-01

**Originality:** Very Good
**Technical Quality:** Very Good
**Clarity Of Presentation:** Very Good
**Impact:** 4

**Recommendation:**

Strong Accept: I recommend accepting the paper and will argue for my recommendation even if other reviewers hold a different opinion.

**Summary:**

This paper presents a novel learning framework for quadruped to learn coordinated dynamic manipulation and locomotion tasks. Instead of training two separate policies for manipulation and locomotion tasks, the proposed approach learns a single unified policy for arm and leg joints motion. The authors use advantage mixing to reduce the coupling between manipulation and locomotion tasks, thereby improving the learning efficiency. Moreover, the authors propose a novel Regularized Online Adaptation algorithm (similar to RMA) to improve the sim-to-real transfer performance. The proposed framework was in simulation and real-world experimentations through several tests. The results show improved learning efficiency, operation capability, and success rates over baseline approaches.

**Issues:**

- Some additional details regarding learning efficiency can be provided.
- How did the difference in PD tracking in simulation and hardware being addressed?
- There are some minor typos. Please carefully proofread the paper.

**Quality Of The Limitations Section:**

Additional details required

**Reviewer Expertise:**

4: The reviewer is confident but not absolutely certain that the evaluation is correct

**Robotics Focus:**

Sufficient demonstration on hardware

**Strengths And Weaknesses:**

### Strength

- It uses advantage mixing to improve the learning efficiency of coupled tasks without compromising on traditional decoupled loco-manipulation approaches.
- This paper shows that coordinating arm manipulation and leg locomotion could improve the robot’s operation space and overall stability.
- The Regularized Online Adaptation algorithm appears to provide good sim-to-real performance.
- The authors clearly explained their simulation and experimental setups and results.
- The trained neural-network policy can realize faster motion when compared to online optimal control (MPC-IK).

### Weakness:

- The dimension of the inputs of the unified policy network is large, which would potentially lead to a very large neural network. While this is a common approach in many end-to-end learning for robotics, it may lead to sampling inefficiency and extended training time. How long does it take to train a policy in simulation?
- The policy outputs joint position commands and then uses PD controllers to track the target positions. However, the joint tracking performance in simulation and robot hardware could be significantly large due to model mismatch, potentially introducing a large sim-to-real gap. How did you account for the PD tracking errors in experiments?

**Summary Of Recommendation:**

Overall, the paper is well written and the results are interesting. While the robot does not move as smooth as SOTA robots in experiment, it is understandable considering the relatively cheap robotic arm being used. The authors presented a comprehensive framework to learn coordinated loco-manipulation policy with an ability to bridge the sim-to-real transfer gap. While some additional details (such as training time, etc.) can be provided, the work presented would provide an excellent example of the advantages of coordinated loco-manipulation policies in legged robots. I highly recommend to accept the paper to CoRL.

---

> ### Author Response · Authors · 2022-08-20
> **Response to Reviewer # 3ot8 (with new experiments) [part 1/2]**
>
> Dear Reviewer,
>
> Thank you for the insightful feedback! We provide clarifications for your concerns below and include new experiments as suggested by other two reviewers. We hope that these answers (+ new experiments) would help increase your confidence. Please let us know if there are any more questions.
>
> > *The dimension of the inputs of the unified policy network is large, which would potentially lead to a very large neural network. While this is a common approach in many end-to-end learning for robotics, it may lead to sampling inefficiency and extended training time. How long does it take to train a policy in simulation?*
> - Indeed, the dimension of the input to the unified is the concatenation of observations from arm/quadruped, and hence, much larger than a standalone quadruped or a standalone robot arm. Note that we provided the training details in Section D of the supplementary material. To reiterate, the policy is trained for 2 billion control steps in IsaacGym using PhysX, where 5000 environments run in parallel. Due to GPU-based simulation, full training takes only 9hrs.
> - Note that these 2 billion steps are similar to what prior works train for just quadruped alone. The reason our unified policy trains in a similar amount of time as a quadruped because our proposed Regularized Online Adaption eliminates the need for two-phase sim2real training common across most end-to-end learning for robotics, especially locomotion [e.g. 22, 23, 24, 25 in the paper].
>
> > *How did the difference in PD tracking in simulation and hardware being addressed?*
> > *The policy outputs joint position commands and then uses PD controllers to track the target positions. However, the joint tracking performance in simulation and robot hardware could be significantly large due to model mismatch, potentially introducing a large sim-to-real gap. How did you account for the PD tracking errors in experiments?*
> - When training the unified policy in simulation, we randomized the environment parameters that are passed into the privileged information encoder to obtain latent extrinsics vector ($z$). In Table 3, we listed the randomization ranges of the environment parameters. In particular, the randomization in the arm motor strength and the leg motor strength specifically compensate for the sim2real gap for PD controllers.
> - To explain concretely, the gains used by the PD controllers ($K_p^{\mathrm{leg}} = 50, K_d^{\mathrm{leg}} = 1, K_p^{\mathrm{arm}} = 5, K_d^{\mathrm{leg}} = 0.5$) are multiplied by uniformly randomly sampled motor strength factors from the training ranges to result in final gains used in the simulator.
> - To highlight an interesting detail, note the randomization range for the arm motor strength is larger than the one for the leg motor strength. This is because the arm is very low-cost and thus arm motors are much less reliable.
>
> > *Some additional details regarding learning efficiency can be provided.*
> > *There are some minor typos. Please carefully proofread the paper.*
> - Thank you for the suggestions. We will do a careful pass to clarify the efficiency discussion and thoroughly address the typos.
>
>
> [1/2] ... continued below...

---

> > ### Author Response · Authors · 2022-08-20
> > **Response to Reviewer # 3ot8 (with new experiments) [part 2/2]**
> >
> > Below are the new experiments we did for rebuttal upon request from other reviewers. We hope you find them useful as well since they further justify our approach.
> >
> > ### New experiment: more accurate visual tracking by averaging poses of two April Tags
> > - We tested the 2-tag method proposed by Reviewer bkQw to capture the track more reliably so that the results can purely focus on the whole-body control performance. We will add this into the camera ready. The result video is here: [https://drive.google.com/file/d/1QsgC7horF8xsTwuBFOO3Z7zujCkepTj3/view?usp=sharing](https://drive.google.com/file/d/1QsgC7horF8xsTwuBFOO3Z7zujCkepTj3/view?usp=sharing)
> >
> > ### More experiments for 'hard' visual tracking setting
> > - We added new experiments suggested by Reviewer KgX5 to include 4 more difficult settings (i.e. target points) for the hard visual tracking instead of one in the paper (note that there are already 10 trials per point).
> > - One of these 4 points is far away from the robot. Even though MPC+IK is not able to reach it with limited workspace, our method can successfully grasp the cup far away because workspace gets increased due to joint coordination.
> > - Other 3 points contain points around the robot that are closer to the robot body that require whole-body coordination to bend the feet for the arm gripper to reach.
> > - Result videos: [https://drive.google.com/file/d/1-w8WTo4WISpMAaMniUBjs7YqsvTO3FLx/view?usp=sharing](https://drive.google.com/file/d/1-w8WTo4WISpMAaMniUBjs7YqsvTO3FLx/view?usp=sharing).
> > - For each of thes 4 new settings (points), we tested 10 trials with the same metric as in Table 7 in the paper as follows:
> >
> > | Hard Ground Points ($p^\mathrm{end}$)| Methods | Success Rate ($\uparrow$) | TTC ($\downarrow$) | IK Failure Rate ($\downarrow$) | Self Collision Rate ($\downarrow$) |
> > | - | - | - | - | - | - |
> > | (0.72, -0.51, 0.34), (0.55, -0.75, -0.43), (0.56, -0.73, 0.5), (0.45, -0.74, 1.80), (0.45, -0.76, -1.8) | Ours | 0.8 | 5.6 | \- | 0 |
> > | ^ | MPC+IK| 0.1 | 22 | 0.2 | 0.5 |
> >
> >
> > [2/2]

---

### Author Response · Authors · 2022-08-28
**[To All] Brief Summary of Rebuttal**

We thank the reviewers for their valuable feedback! We are glad that reviewers liked the proposed learning framework (Reviewer 3ot8 & Hcpy & bkQw & KgX5), found simulation experiments and real-world robot evaluations extensive (Area Chair A5R8), impressive (Reviewer Hcpy & bkQw) and clear (Reviewer 3ot8). We are pleased to report that we finished all the experiments that the reviewers suggested. We made detailed responses directly to each review. Here, we recap only the main contributions of this paper and the results from the additional experiments.

### Key contributions of this paper
1. We presented a custom-built budget hardware setup (6k USD) and a new whole-body control pipeline for a low-cost fully-untethered legged manipulator using only onboard computation and onboard power.
2. We proposed a method to learn one unified policy to control and coordinate all joints of the legs and the arm by exploiting the causal structure with respect to manipulation and locomotion to stabilize and speed up learning, and by adding regularization to domain adaptation in an online fashion to bridge the Sim2Real gap for high-dim observation and control.
3. Our method is compatible with 3 operating modes: teleoperation, visual tracking, and open-loop demonstration replay, which are tested extensively in real-world experiments.

$\newline$

Below are the experiments we added for rebuttal upon suggestions from the reviewers. We hope you find them useful as well since they further justify our approach.

### New experiment: more accurate visual tracking by averaging poses of two April Tags
- We tested the 2-tag method proposed by Reviewer bkQw to capture the track more reliably so that the results can purely focus on the whole-body control performance. We will add this into the camera ready. The result video is here: [https://drive.google.com/file/d/1QsgC7horF8xsTwuBFOO3Z7zujCkepTj3/view?usp=sharing](https://drive.google.com/file/d/1QsgC7horF8xsTwuBFOO3Z7zujCkepTj3/view?usp=sharing)

### More experiments for 'hard' visual tracking setting
- We added new experiments suggested by Reviewer KgX5 to include 4 more difficult settings (i.e. target points) for the hard visual tracking instead of one in the paper (note that there are already 10 trials per point).
- One of these 4 points is far away from the robot. Even though MPC+IK is not able to reach it with limited workspace, our method can successfully grasp the cup far away because workspace gets increased due to joint coordination.
- Other 3 points contain points around the robot that are closer to the robot body that require whole-body coordination to bend the feet for the arm gripper to reach.
- Result videos: [https://drive.google.com/file/d/1-w8WTo4WISpMAaMniUBjs7YqsvTO3FLx/view?usp=sharing](https://drive.google.com/file/d/1-w8WTo4WISpMAaMniUBjs7YqsvTO3FLx/view?usp=sharing).
- For each of thes 4 new settings (points), we tested 10 trials with the same metric as in Table 7 in the paper as follows:

| Hard Ground Points ($p^\mathrm{end}$)| Methods | Success Rate ($\uparrow$) | TTC ($\downarrow$) | IK Failure Rate ($\downarrow$) | Self Collision Rate ($\downarrow$) |
| - | - | - | - | - | - |
| (0.72, -0.51, 0.34)\(0.55, -0.75, -0.43)\(0.56, -0.73, 0.5)\(0.45, -0.74, 1.80)\(0.45, -0.76, -1.8) | Ours | 0.8 | 5.6 | \- | 0 |
| ^ | MPC+IK| 0.1 | 22 | 0.2 | 0.5 |

---

### Meta-Review · Area_Chair_A5R8 · 2022-08-10

**Recommendation:** Accept (Oral)
**Confidence:** 5

**Metareview:**

This paper presents a reinforcement learning framework for learning mobile manipulation using a quadruped with an arm. The main technical contributions are 1) a curriculum learning with only one parameter (Advantage Mixing) and 2) regularized online adaptation for sim-to-real transfer. The trained policy is evaluated extensively in both simulation and hardware experiments.

The paper is well written and the contributions are clear. The reviewers were very positive and raised only some presentation issues, which the authors successfully addressed in the rebuttal. They also conducted additional visual tracking experiments with better target localization accuracy to further reinforce the efficacy of their method.

**Best Paper Nomination:**

Yes

---

> ### Author Response · Authors · 2022-08-21
> **Response to Area Chair # A5R8 (with new experiments) [part 1/2]**
>
> Dear AC,
>
> We thank you and all the reviewers for the insightful feedback and for appreciating our thorough real-world results, the novelty and effectiveness of our method, and detailed ablation studies in simulation. We have clarified all the reviewers' questions in direct responses and post a summary response separately as well. We are pleased to report that we finished all the experiments that the reviewers suggested.
>
> > *An additional comment from the Area Chair is that [9,10] do not completely decouple the base (leg) and arm motions and do realize coordinated motions. In [9], for example, a whole-body controller is used to compute all joint torques simultaneously given the end-effector motion and foothold that are determined based on operator inputs. Therefore, the MPC+IK method used as baseline is not the SOTA of model-based approaches.*
> - We thank you for pointing out the correction for [9,10]. Both of these prior works are stellar demonstrations of legged manipulation and are sources of inspiration for us. We will modify our wordings in the introduction section to better reflect the fact that [9,10] do not completely decouple the base (leg) and arm motions and do realize coordinated motions.
> - However, there are significant differences in [9,10] from our proposed approach. [9] relies on the model-based optimization methods for whole-body control, whereas our focus is learning-based approach and not assuming much a priori information about the robots. [10] does not control the leg joints directly but relies on high-level command API provided by Spot robot from Boston Dynamics, whereas our method controls every joint of the arm and legs directly in 50Hz.
> - We attempted to compare these SOTA [9,10] model-based optimization controllers but *the code was not released* and it was very hard for us to reproduce the results given huge differences in the hardware setups. For example, both setups in [9, 10] are on quadrupeds of large form factors and expensive motors (of the order of 100k USD), but in contrast, our setup only 6K USD total and is fully untethered unlike [10]. As a result, the onboard computation power (1 Raspberry Pi + 1 Nvidia Jetson) is not enough to support the real-time solving of optimization problems formulated in [9, 10] (if the code was released).
> - Hence, we choose the MPC+IK as the real-world baseline, even though we are aware that it’s not the SOTA model-based controller (note we do not claim it is SOTA in the paper either).
> - Interestingly, this points to another advantage of using a single unified network as it only needs a single forward pass of a network to compute all target joints at once, and hence, runs efficiently and stably on low-compute hardware.
>
> [1/2] ...continued below...

---

> > ### Author Response · Authors · 2022-08-21
> > **Response to Area Chair # A5R8 (with new experiments) [part 2/2]**
> >
> > Below are the new experiments we did for rebuttal upon request from other reviewers. We hope you find them useful as well since they further justify our approach.
> >
> > ### New experiment: more accurate visual tracking by averaging poses of two April Tags
> > - We tested the 2-tag method proposed by Reviewer bkQw to capture the track more reliably so that the results can purely focus on the whole-body control performance. We will add this into the camera ready. The result video is here: [https://drive.google.com/file/d/1QsgC7horF8xsTwuBFOO3Z7zujCkepTj3/view?usp=sharing](https://drive.google.com/file/d/1QsgC7horF8xsTwuBFOO3Z7zujCkepTj3/view?usp=sharing)
> >
> > ### More experiments for 'hard' visual tracking setting
> > - We added new experiments suggested by Reviewer KgX5 to include 4 more difficult settings (i.e. target points) for the hard visual tracking instead of one in the paper (note that there are already 10 trials per point).
> > - One of these 4 points is far away from the robot. Even though MPC+IK is not able to reach it with limited workspace, our method can successfully grasp the cup far away because workspace gets increased due to joint coordination.
> > - Other 3 points contain points around the robot that are closer to the robot body that require whole-body coordination to bend the feet for the arm gripper to reach.
> > - Result videos: [https://drive.google.com/file/d/1-w8WTo4WISpMAaMniUBjs7YqsvTO3FLx/view?usp=sharing](https://drive.google.com/file/d/1-w8WTo4WISpMAaMniUBjs7YqsvTO3FLx/view?usp=sharing).
> > - For each of thes 4 new settings (points), we tested 10 trials with the same metric as in Table 7 in the paper as follows:
> >
> > | Hard Ground Points ($p^\mathrm{end}$)| Methods | Success Rate ($\uparrow$) | TTC ($\downarrow$) | IK Failure Rate ($\downarrow$) | Self Collision Rate ($\downarrow$) |
> > | - | - | - | - | - | - |
> > | (0.72, -0.51, 0.34), (0.55, -0.75, -0.43), (0.56, -0.73, 0.5), (0.45, -0.74, 1.80), (0.45, -0.76, -1.8) | Ours | 0.8 | 5.6 | \- | 0 |
> > | ^ | MPC+IK| 0.1 | 22 | 0.2 | 0.5 |
> >
> > $\newline$
> >
> > #### Reference
> > [9] C. D. Bellicoso, et al. Alma-articulated locomotion and manipulation for a torque-controllable robot. ICRA 2019.
> > [10] S. Zimmermann, et al. Go fetch!-dynamic grasps using boston dynamics spot with external robotic arm. ICRA 2021.
> >
> > $\newline$
> >
> > [2/2]